# Facile Label-Free Electrochemical DNA Biosensor for Detection of Osteosarcoma-Related Survivin Gene

**DOI:** 10.3390/bios12090747

**Published:** 2022-09-09

**Authors:** Yao Chen, Yu Zhong, Ji-Xing Ye, Yun Lei, Ai-Lin Liu

**Affiliations:** Department of Pharmaceutical Analysis, Higher Educational Key Laboratory for Nano Biomedical Technology of Fujian Province, The School of Pharmacy, Fujian Medical University, Fuzhou 350122, China

**Keywords:** electrochemical DNA biosensor, celestine blue, hybridization indicator, osteosarcoma, survivin gene

## Abstract

A sensitive and selective electrochemical deoxyribonucleic acid (DNA) biosensor was developed for the determination of a osteosarcoma-related survivin gene by using celestine blue (CB) as a label-free hybridization indicator. The proposed strategy adopted a facile and low-cost working electrode with no need for other substances for electrode or DNA functionalization. The interaction mode between CB and DNA was studied by electrochemical and spectroscopic approaches, illustrating that the possible mode was intercalation with a binding number of 2 and a binding constant *β* of 10^12.87^. Moreover, the label-free electrochemical DNA biosensor exhibited a good linear relationship toward the target gene in a range from 1.00 nM to 50.00 nM with a detection limit as low as 0.046 nM using 3*σ* estimating system. This facile and low-cost electrochemical method realized the rapid detection and accurate quantification of the target sequence in complicated serum samples, endowing its promising potential in the diagnosis and monitoring of genetic diseases.

## 1. Introduction

Osteosarcoma, originating from primitive bone-forming mesenchymal cells, is the most common malignant bone tumor, which is characterized by the production of highly malignant spindle stromal cells [1]. Among adolescence tumors, osteosarcoma ranks eighth in general incidence (2.4%). According to the statistics, the overall 5-year survival rate with respect to osteosarcoma is about 68% [2]. For patients accompanied with other metastatic diseases, regrettably, the 5-year survival rate is below 30% [3]. In this regard, osteosarcoma is still a highly invasive and severely lethal disease. It has been documented that the survivin gene is closely relevant to the pathogenesis of osteosarcoma [4]. Additionally, the survivin gene can be employed as a biomarker for the prognosis of osteosarcoma, abating the shortage that prognosis is mainly determined by the merely assistant prognostic parameters concerning chemotherapy, tumor size and surgical margin width [5]. Moreover, the survivin gene has the potential to provide a clinically therapeutic alternative for cancer-directed treatment [6]. Hence, the determination of the survivin gene is able to afford the early clinical diagnosis and timely treatment of osteosarcoma, which can greatly improve patients’ survival rate and even their quality of life to a large extent.

Progress has been made by means of detecting genomic characterizations of available samples in clinical practice, such as flow cytometry (FC), chromosome analysis, fluorescence in situ hybridization (FISH), real-time polymerase chain reaction (RT-PCR) and so on [7]. However, these strategies may be limited more or less by high cost, time consumption and inaccuracy. Therefore, the development of a simple, rapid, highly sensitive and selective approach for the early diagnosis of disease has broad application prospects. The electrochemical method would be a good choice due to its merits of easy operation, rapid response, low detection limit and facile modification [8,9]. Recently, electrochemical deoxyribonucleic acid (DNA) sensors have drawn widespread attention in biochemical analysis by discerning DNA recognition events and transforming them into electrochemical signal. There are several corresponding reports concerning the direct DNA oxidation experiment at solid electrodes (such as gold, glassy carbon, carbon paste, and so on) or via other modes [10]. Although the direct oxidation method improves the reliability of determination, the application of such a mode is limited due to the relatively small oxidation signal of purine bases [11]. To address this issue, equipping electroactive indicators to electrochemical sensors is greatly applied to monitoring DNA hybridization events. For instance, Millan et al. [12] firstly proposed that small electroactive molecules were adopted to discriminate single-stranded DNA (ssDNA) and double-stranded DNA (dsDNA), then investigated the interaction mode of the [Co(Phen)_3_]^2+^ hybridization indicator with the corresponding nucleic acid. 

Depending on its binding mode to the fixed DNA, hybridization indicators can be divided into two types: labeled and label-free modes. Between them, the labeled mode mainly refers to some inorganic or biological materials with electrochemical activity or catalytic activity (such as ferrocene and horseradish peroxidase), which are fixed on the gene fragments by adsorption, covalent bonding, or biological affinity. Then, the hybridization process is indicated by detecting the electrical signals of the catalytic substrates [13]. Despite the high sensitivity of this method, the detection process involves synthesis, labeling, separation, purification, etc., which is complicated, time-consuming, and easily causes sample loss in the operation process [14].

Label-free hybridization indicators are a class of electrochemically active compounds whose redox potential lies between the electrochemical windows of DNA, and these objects can interact with ssDNA and dsDNA in different ways [15]. When redox reaction occurs, the indicator can exchange electrons with DNA and complete electrochemical redox reaction on the electrode surface through long-distance electron transport process. Based on the difference of a selective binding ability between the indicator and ssDNA or dsDNA, the DNA hybridization process is manifested by the measurable signals (current, potential, electric resistance, etc.) [16]. Therefore, the label-free strategy, allowing for the direct translation of the recognition behavior into the readable signal in a real-time manner, is still one of the mainstream choices in the electrochemical DNA sensor as clinical diagnosis tool because of simple operation and rapidness. Label-free hybridization indicators mainly include metal complexes and organic dyes. For instance, Tansil et al. [17] synthesized PIND-[Os(bpy)_2_Cl]^+^, Ru(bpy)_2_dppz, Co(bpy)_3_(ClO_4_) and Os(bpy)_3_Cl_2_ as hybridization indicators by utilizing Co, Ru, pyridine and so forth to detect target DNA with high sensitivity and selectivity. Celestine blue (CB) as a simple compound with positive charge is widely applied to fabricate the hydrogen peroxide or glucose sensors [18]. So it can be used as a promising hybridization indicator via binding to nucleic acids by electrostatic attraction, which can be employed in electrochemical DNA biosensors. Bazgir et al. [19] designed the CB-modified carbon paste electrode and investigated the interaction mode between CB and nucleic acid through spectroscopic and electrochemical approaches. The detection limit of this approach was as low as 0.48 nM. Herein, CB was chosen as the hybridization indicator for the measurement of target DNA to construct a highly sensitive and selective detection strategy owing to its high electron transfer efficiency.

In this study, a facile and sensitive label-free electrochemical DNA biosensor was constructed to determine osteosarcoma-related the survivin gene by employing CB as label-free hybridization indicator. The fabrication procedure of the sensor was displayed in Figure 1. Upon hybridization of the capture probe with target DNA, CB intercalated into dsDNA and formed a new complex, leading to a slide in oxidative peak current produced from the hybridization indicators. Under the optimized conditions, the prepared label-free biosensor demonstrated a linear relationship with the concentration of target DNA ranging from 1.00 nM to 50.00 nM and a detection limit as low as 0.046 nM. Furthermore, the proposed sensor displayed great selectivity in measuring a disease-related genetic marker in 10% human serum samples containing interfering sequences, demonstrating its potential application prospect in rapid diagnosis as well as long-term prognosis monitoring.

## 2. Experimental Sections

### 2.1. Reagents and Apparatus

CB was obtained from Sinopharm Chemical Reagent Co., Ltd. (Shanghai, China). N-(3-dimethylaminopropyl)-N′-ethylcarbodiimide hydrochloride (EDC) and N-hydroxysuccinimide (NHS) were obtained from Sigma-Aldrich (St. Louis, MO, USA). Fish sperm double-stranded DNA was purchased from Fluka Inc. (Shanghai, China). Other reagents were analytical reagent (AR) grade, and all solutions in this assay were prepared with Milli-Q water (18.3 MΩ·cm resistivity). The corresponding buffer solutions were as follows: phosphate buffer solution (PBS) was prepared by 50 mM NaH_2_PO_4_-Na_2_HPO_4_ and 20 mM NaCl, then 50 mM H_3_PO_4_ and 50 mM NaOH solution were used to adjust pH value; tris-ethylenediaminetetraacetic acid buffer (TE) used to dissolve the oligonucleotides involved 10 mM Tris-HCl (pH 8.0), 1.0 M ethylenediaminetetraacetic acid (EDTA) and 0.1 M NaCl; the dilution buffer solutions for target sequences and DNA probes were 10 mM Tris-HCl (pH 7.0) and acetate buffer (pH 4.8), respectively. All the oligonucleotides (shown in Table 1) were synthesized and purified by Sangon Biological Engineering Technology and Services Co., Ltd. (Shanghai, China). Electrochemical experiments were performed with a CHI 1030A electrochemical analyzer (CH Instruments, Shanghai, China) equipped with a conventional three-electrode system consisting of a glassy carbon electrode (GCE, 3.0 mm in diameter, acted as working electrode), a platinum wire (worked as counter electrode) and an Ag/AgCl electrode (3.0 M KCl, served on reference electrode). Ultraviolet–visible (UV–vis) absorption spectrum was performed on a UV-2450 spectrophotometer (Shimadzu, Kyouto, Japan).

### 2.2. Interaction of CB with dsDNA in Solution

In total, 10 μL of 1.0 nM CB solution was added in 1 mL of PBS (pH 7.0). Then, various concentrations of salmon sperm dsDNA were added into the above prepared solutions. The final concentration of CB was 10 μM. The concentrations of salmon sperm dsDNA were 10, 20, 50, 75 and 100 mg/L, respectively. The cyclic voltammogram (CV) curve was performed with the potential in a range from 0 V to 1.2 V at the scan rate of 100 mV/s, and differential pulse voltammetry (DPV) curve was carried out ranging from 0.4 V to 1.0 V with a scan rate of 100 mV/s.

### 2.3. UV-vis Spectroscopy Experiment 

A mixed solution of CB and salmon sperm dsDNA at a certain concentration was added into a 2 mL EP tube and diluted with PBS (pH 7.0). The UV-vis absorption spectrum was detected once the mixture incubated for 0.5 h at room temperature.

### 2.4. Fabrication of Electrochemical DNA Biosensor for Survivin Gene Detection 

The clean GCE surface was firstly polished with 0.3 mm and 0.05 mm Al_2_O_3_ slurry in order, and treated by ultrasonic wash with 50% nitric acid (volume ration of Milli-Q water vs. HNO_3_ = 1:1), ethanol as well as Milli-Q water, successively. Then, the prepared counterpart was oxidized at the constant potential of 0.5 V in PBS (pH 7.4) for 1 min, cleaned with Milli-Q water and dried with nitrogen. Subsequently, the pretreated GCE was immersed into 50 mM PBS (pH 7.4, 100 μL) involving 5 mM EDC as well as 8 mM NHS for 0.5 h, followed by washed with Milli-Q water to remove redundant EDC and NHS on the surface of electrode. After that, 10 μL of 1 μM ssDNA was added onto the electrode surface followed by air-dried to obtain ssDNA/GCE, and ssDNA/GCE was rinsed with Tris-HCl buffer solution (pH 7.0) to remove the non-specific binding probe, followed by immersing in Tris-HCl buffer involving complementary ssDNA at a certain concentration at 45 °C for 1 h to obtain dsDNA/GCE.

## 3. Results and Discussion

### 3.1. Electrochemical Behavior of CB

CB is a kind of cationic indicator, and the molecular structure is shown in Figure 2. Figure 1A shows the CV curve of CB in PBS (pH 7.0). There was an excellent peak shape at the potential of +0.759 V, indicating that CB possessed good electrochemical activity on GCE. To further investigate the electrocatalytic activities of CB in PBS (pH 7.0), CV curves of CB were examined at different scan rates (*v*). It was observed that the oxidation peak current of CB was significantly enhanced and shifted to more positive potential with the increment of *v* (Figure 1B). The oxidation peak current (*I*_pa_) was linear with *v* from 20 to 140 mV/s, obeying the regression equation of *I*_pa_ (μA) = 0.07 *v* (mV/s) + 9.06 (R^2^ = 0.9956), implying a diffusion-controlled process on the surface of electrode (Figure 1C).

For studying the impact of pH on the current signal of CB, the corresponding CV curves in 50 mM PBS with different pH were obtained. As the pH value increased, the oxidation peak current of CB continuously increased, reached the maximum value at pH 7.0, and then decreased gradually (Figure 2A,B). At the same time, the oxidation peak potential of CB progressively shifted in a negative direction with the enhancement of pH (Figure 2A,C). The oxidation peak potential (*E*^0^) was linear versus pH from 4.0 to 10.0, demonstrating that hydrogen ions (H^+^) are involved in the electrochemical reaction (Figure 2C), that is, the current signal of CB decreased sharply when pH value was more than 7.0 (Figure 2B). Thus, pH 7.0 was chosen as the optimal acidity in this work.

### 3.2. The Interaction between CB and Salmon Sperm dsDNA

#### 3.2.1. Electrochemical Investigation on the Interaction between CB and Salmon Sperm dsDNA

In order to investigate the practical application of CB for the measurement of DNA, an electrochemical experiment on the interaction between CB and DNA duplex was performed. In Figure 3A, the oxidation peak current gradually decreased, meanwhile, oxidation peak potential slightly shifted to positive potentials with the increment of concentration for salmon sperm dsDNA, implying the formation of a new complex. The result was supported by Carter and Bard’s work, which demonstrated that the enhanced separation of peak potential while a small molecule bounded to nucleic acid via intercalation mode [18]. In theory, intercalative binding between CB and dsDNA forms a new complex without electrochemical activity. The oxidative current showed a negative correlation with the dsDNA concentration (10–100 mg/L, Figure 3B), and the linear regression equation was *I*_pa_ (μA) = −0.09*C*_DNA_ (mg/L) + 14.82, (R^2^ = 0.9928).

#### 3.2.2. UV–vis Spectroscopic Study on the Interaction between CB and Salmon Sperm dsDNA

To further study the possible interaction mode of CB with dsDNA, UV-vis absorption spectrum was recorded on UV-2450 spectrophotometer. The salmon sperm dsDNA had strong absorption peak at 258 nm (Figure 3C, curve a). Two strong absorption peaks respectively located at 262 nm and 635 nm for CB (Figure 3C, curve b). The maximum absorption peak (635 nm) of CB was shifted to 639 nm, and the absorbance was reduced significantly after interaction of CB with dsDNA for 30 min (Figure 3C, curve c). The results were consistent with the interpretation that the UV spectrum of the small molecules appears hypochromic effect and red shift when intercalating with dsDNA [20]. These two obvious spectral effects were attributed to a strong interaction between the electronic state of the intercalating chromophore and that of the DNA bases. This facilitates a decrease in the energy of π-π* electron transition, which reflected in the observed red shift. At the same time, the empty π* orbital is partially filled by electrons, and this contributes to the hypochromic effect. Therefore, it can be inferred that the binding mode between CB and salmon sperm dsDNA was the specific intercalation mode.

### 3.3. The Equilibrium Constant between CB and Salmon Sperm dsDNA

The equilibrium constant between CB and salmon sperm dsDNA was studied based on the assumption that only one simple complex DNA-*n*CB was formed [21]. The binding number (*n*) and equilibrium constant (*β*) can be calculated by the following equations:DNA + *n*CB ⇌ [DNA-*n*CB](1)

The equilibrium constant *β* can be expressed as follows:(2)$$\beta =\frac{\left[DNA\text{-}nCB\right]}{\left[DNA\right]{\left[CB\right]}^{n}}$$

The following equation can be further inferred:(3)$$\frac{1}{\Delta I}=\frac{1}{\Delta I_{\max}}+\frac{1}{\beta \Delta I_{\max}{\left[\mathrm{CB}\right]}^{n}}$$

The binding number (*n*) can be inferred from Equation (4):(4)$$\mathrm{lg}\left(\frac{\Delta I}{\Delta I_{\max}-\Delta I}\right)=\mathrm{lg}\left(\beta \right)+n\mathrm{lg}\left[CB\right]$$
where Δ*I*_max_ is standing for the peak current of the salmon sperm dsDNA, Δ*I* is standing for the peak current of the dsDNA-*n*CB complex. According to the calculation formula above, a linear graph between lg[Δ*I*/(Δ*I*_max_ − Δ*I*)] and lg[CB] was made as shown in Figure 3D, where the slope is binding number *n* and the intercept is lg*β*. The *n* = 2.24 and *β* = 10^12.87^ could be calculated from Figure 3D, which indicated that only one intercalation compound (DNA-2CB) was formed during the whole inserting process, and the binding ratio for CB and dsDNA was 2:1.

### 3.4. Electrochemical Characterization of Modified GCE Electrodes

Electrochemical impedance spectroscopy (EIS) is usually applied to explore the interfacial properties of fabricated electrodes and the electron transfer resistance. The diameter of the semicircle portion is linear with the electron transfer resistance (*R*_et_), and the growth of diameter reflects enhancement of interfacial *R*_et_. As shown in Figure 4, the EIS data were recorded in 0.01 M K_3_[Fe(CN)_6_]/K_4_[Fe(CN)_6_] (1:1) solution involving 0.1 M KCl. The bare GCE presented the smallest semicircle (curve a), illustrating a fast electron transfer process. After accumulation of CB onto the GCE, the resistance improved from 45 Ω to 145 Ω (curve b). The resistance value was enlarged to 669 Ω of curve c, which was caused by electrostatic repulsion between ssDNA S1 with negatively charged phosphate backbone and Fe(CN)_6_^3−/4−^. The radius of curve d amplified about two-fold, indicating a slow electron-transfer between ssDNA and the electrode surface as a result of the enrichment of CB. The *R*_et_ was further expanded after the formation of dsDNA on the GCE surface (curve e). The value of *R*_et_ was broadened to 2846 Ω with the addition of CB, which could be explained by the enrichment of CB on the electrode surface and the intercalation proceed (curve f).

### 3.5. Performance of the Designed Biosensor for Osteosarcoma-Related Survivin Gene Detection

DPV was used to evaluate the recognition capability of the biosensor by detecting three different kinds of DNA sequences with the ssDNA/GCE, including target DNA (S2), single-base mismatched DNA (S3) and non-complementary DNA (S4); all the sequences were at the same concentration of 100 nM, the potential ranged from 0.4 V to 1.0 V at a scan rate of 100 mV/s (Figure 5A). As shown in Figure 5A, compared with the other two base-mismatch sequences (curve b and c), the minimum peak current was obtained for target DNA (S2) (curve d), indicating the probe immobilized on GCE was hybridized with the correspondingly complementary sequence. The peak current different (Δ*I*) was obtained for S1/GCE after hybridization with different mismatched sequences (the inset in Figure 5A). When interacted with non-complementary DNA, there was almost no peak current change. A slight decrease in peak current was observed when S3 interacted with the ssDNA/GCE which indicating that the complete hybridization was not accomplished due to the one base mismatched.

This biosensor showed an effective discrimination capacity in gene detection for clinical diagnosis. Human serum, as a highly complex biological fluid, possesses plenty of proteins and a variety of small biomolecules. It can be seen that the peak current difference for the hybridization with the complementary sequence S2 in 10% human serum sample (diluted with PBS) was 1.74-fold higher than in PBS, ascribed to the blocking of the proteins existing in serum for the GCE surface (the inset in Figure 5B). To further investigate the interference produced by coexisting protein and other small molecules, the specificity of the proposed biosensor was assessed in human serum. Different DNA sequences, including S3, S4, S5 (partial BCR sequence) and S6 (partial ABL sequence), were added with the same concentration as S2 into human serum as interference background (S5 and S6 are the partial BCR/ABL fusion gene correlated to chronic myelogenous leukemia). The results exhibited that current signal of S2 was not affected by the mismatched sequences, indicating the sensor could perfectly distinguish the target from various interference sequences (Figure 5C). Compared with the Bazgir’s work [19], our further study in the sequence selectivity and anti-interference experiments showed its great potential in clinical diagnosis.

Furthermore, under optimal conditions, the current decreased linearly with the S2 concentration ranging from 1.00 nM to 50.00 nM (Figure 5D). The regression equation was Δ*I* (μA) = 0.56 + 0.38 lg*C*_DNA_ (nM) with R^2^ = 0.9841 and the detection limit of 0.046 nM (S/N = 3), demonstrating that the sensor possessed good sensitivity even in human serum. Thereinto, Δ*I* was the difference in the peak current before and after hybridization with the complementary target. The quantitative determination of osteosarcoma-related gene was successfully achieved. A comparison of the different methods applied for the survivin gene determination is presented in Table 2. It is seen that the proposed biosensor can provide higher sensitivity. Additionally, it is a promising platform for survivin-gene screening in the early diagnosis of cancer.

## 4. Conclusions

Herein, a label-free electrochemical DNA biosensor was constructed for detecting osteosarcoma-related the survivin gene in human serum samples, using CB as the DNA hybridization indicator. The biosensor was prepared for the purpose of evaluating the survivin gene in serum sample. The proposed biosensor presented the salient merits of a low-cost, simple operation process and rapid response. Moreover, it exhibited great selectivity and anti-interference capacity for the measurement of the survivin gene in a serum sample containing various mismatched sequences as an interference substance. Additionally, serum acted as a block agent for the GCE surface, resulting in a surge in the peak current change upon hybridizing with the target sequence, further improving the sensitivity and accuracy of the biosensor. We believed that the developed label-free electrochemical DNA biosensor can serve as a potential candidate for the sensitive and accurate determination of the survivin gene in complicated serum milieu, which is promising for the clinical diagnosis of osteosarcoma. This presented biosensor exhibited potential clinical applications for the early diagnosis of osteosarcoma.

## Data Availability

Not applicable.

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
