# Peer review of "Facile Label-Free Electrochemical DNA Biosensor for Detection of Osteosarcoma-Related Survivin Gene"

_biosensors, 2022, doi:10.3390/bios12090747_

Round 1

Reviewer 1 Report

This paper studied a label free electrochemical DNA biosensor for the detection of osteoma genes. The overall logic of the article is clear and the data is sufficient. There are some problems that need to be solved before it is considered for publication. I believe it can be published after minor revision.

For example, why CB and DNA combine into intercalation compounds needs to be fully explained. Additionly,in paper line 171 "as the PH value increased" a, there is ambiguity mismatching between the “increased” and the data display.Please consider replacing them with more appropriate sentences.

Author Response

This paper studied a label free electrochemical DNA biosensor for the detection of osteoma genes. The overall logic of the article is clear and the data is sufficient. There are some problems that need to be solved before it is considered for publication. I believe it can be published after minor revision.

Comment 1: Why CB and DNA combine into intercalation compounds needs to be fully explained?

Answer: Thanks for the reviewer’s kindly comment on our manuscript. We supplemented the corresponding section that “The maximum absorption peak (635 nm) of CB was shifted to 639 nm and the absorbance was reduced significantly after interaction of CB with dsDNA for 30 min (Fig. 3C, curve c). These results were consistent with the interpretation that the UV spectrum of the small molecules appears hypochromic effect and red shift when intercalating with dsDNA [20]. These two obvious spectral effects were attributed to a strong interaction between the electronic state of the intercalating chromophore and that of the DNA bases. This facilitates a decrease in the energy of π-π* electron transition, which reflected in the observed red shift. At the same time, the empty π* orbital is partially filled by electrons, and this contributes to the hypochromic effect. Therefore, it can be inferred that the possible binding mode between CB and salmon sperm dsDNA was the specific intercalation mode.” in the revised manuscript.

Comment 2: Additionly, in paper line 171 "as the pH value increased" a, there is ambiguity mismatching between the “increased” and the data display. Please consider replacing them with more appropriate sentences.

Answer: Thanks for the reviewer’s kindly comment and suggestion on our manuscript. According to reviewer's recommendation, we corrected the corresponding content that “As the pH value increased, the oxidation peak current of CB continuously increased, reached the maximum value at pH 7.0, and then decreased gradually (Fig. 2A and Fig. 2B). ” in the revised manuscript.

Reviewer 2 Report

Dear Editor,

I went through the paper by Chen and co-worker. Although the experimental has been properly designed and described, and the scientific soundness of the result is clear, I don't find enough innovation to support the publication of the paper in its present form.

Indeed, the proposed technique has been introduced several years ago by Bazgir and co-workers (reference 19 in the paper). In the way the results are presented, and also for the kind of tests carried out, the manuscript is similar to Bazgir's work, with the sole "innovation" related to its application to a specific problem (the osteosarcoma-related survivin gene). This means, in practice, that different DNA sequences have been employed, and this is not enough to ensure its publication.

In order to make it more original, I would modify the presentation of the results in order to better focus on the pros of the proposed techniques with respect to the ones employed at the state-of-the-art, briefly cited by authors in the introduction.

This means that the obtained performances in terms of linearity, detection limit, sensitivity and selectivity have to be compared to the one reported in literature for survivin gene detection. In this way, the selectivity experiments (which are original in this paper with respect to Bazgir's manuscript) can be better evaluated with respect to literature.

As any further experiment is needed, I suggest MINOR REVISIONS, and ask authors to better focus their paper on the application in order to enhance its novelty.

Author Response

I went through the paper by Chen and co-worker. Although the experimental has been properly designed and described, and the scientific soundness of the result is clear, I don't find enough innovation to support the publication of the paper in its present form.

Comment 1: Indeed, the proposed technique has been introduced several years ago by Bazgir and co-workers (reference 19 in the paper). In the way the results are presented, and also for the kind of tests carried out, the manuscript is similar to Bazgir's work, with the sole "innovation" related to its application to a specific problem (the osteosarcoma-related survivin gene). This means, in practice, that different DNA sequences have been employed, and this is not enough to ensure its publication.

Answer: Thanks for the reviewer’s kindly comment and suggestion on our manuscript. According to reviewer's recommendation, we added the corresponding section that “On the basis of Bazgir’s work, the aforementioned selectivity and anti-interference results further endowed its great potential in clinical diagnosis.” in the revised manuscript.

Comment 2: In order to make it more original, I would modify the presentation of the results in order to better focus on the pros of the proposed techniques with respect to the ones employed at the state-of-the-art, briefly cited by authors in the introduction.

This means that the obtained performances in terms of linearity, detection limit, sensitivity and selectivity have to be compared to the one reported in literature for survivin gene detection. In this way, the selectivity experiments (which are original in this paper with respect to Bazgir's manuscript) can be better evaluated with respect to literature.

As any further experiment is needed, I suggest MINOR REVISIONS, and ask authors to better focus their paper on the application in order to enhance its novelty.

Answer: Thanks for the reviewer’s kindly comment and suggestion on our manuscript. According to reviewer's recommendation, we added the corresponding content that “A comparison of the different methods applied for survivin gene determination was presented in Table 2. It is seen that the proposed biosensor can provide a higher sensitivity. And it’s a promising platform to survivin gene screening in early diagnosis of cancer.” in the revised manuscript.